# Outcome Prediction in Histopathology: A Multimodal Three-Stage Deep Learning Framework

**Nilanjan Chattopadhyay** [ID]    NILANJAN.CHATTOPADHYAY@AIRAMATRIX.COM
**Saheli Datta**    SAHELI.DATTA@AIRAMATRIX.COM
**Nitin Singhal** [ID]    NITIN.SINGHAL@AIRAMATRIX.COM
*AIRA Matrix, Mumbai, India*

**Editors:** Accepted for publication at MIDL 2025

## Abstract

Accurate outcome prediction is paramount in histopathology for effective cancer management. We present a novel, high-performance multimodal deep learning framework that efficiently integrates information from whole slide images (WSIs) and, optionally, clinical data to significantly enhance prediction. The first stage achieves precise tumor detection using a custom UNet (ConvNeXtv2 encoder for robust segmentation; decoder with residual connections, bottleneck, and SE blocks). To optimize training and generalization, we introduce a strategic patch selection method that enhances generalization. The second stage efficiently extracts highly informative and compressed feature representations from selected regions using a ResNeXt50 network, pre-trained with DINO. The third stage aggregates these features, combines them with clinical parameters (if available), and predicts outcomes via ResNet18. Critically, the framework leverages a multimodal approach, combining WSI image features with clinical parameters for robust outcome prediction. The framework's efficacy is rigorously demonstrated through experiments on biopsy Gleason Grade classification, metastasis, and BRCA2 mutation prediction. Comparative evaluation against Multiple Instance Learning (MIL) approaches highlights superior performance and effective multimodal data utilization.

**Keywords:** Multimodal Learning, Tumor Detection, Outcome Prediction

## 1. Introduction

Accurate prediction of disease progression from tissue samples is crucial for effective cancer treatment. Whole slide images (WSIs), which contain detailed information at the cellular level, are a valuable resource. However, their high dimensionality presents significant analytical challenges. Deep learning offers a promising approach to WSI analysis, but robust tumor identification and effective integration of multimodal data remain key obstacles. Traditional histopathological assessment can be subjective, with potential for inter-observer variability. Computational pathology aims to provide more objective analysis, but the large size of WSIs poses computational difficulties. Patch-based methods, where WSIs are divided into smaller segments, are commonly used, requiring effective patch-selection strategies. Multiple Instance Learning (MIL) (Ilse et al., 2018), a common WSI analysis technique, treats WSIs as bags of patches for overall label prediction. However, MIL can be computationally expensive, often requiring processing of numerous patches without prioritizing the most informative regions, which can limit its efficiency, especially in large-scale studies. To address these limitations, our work introduces a novel multimodal deep learning framework designed for improved outcome prediction.

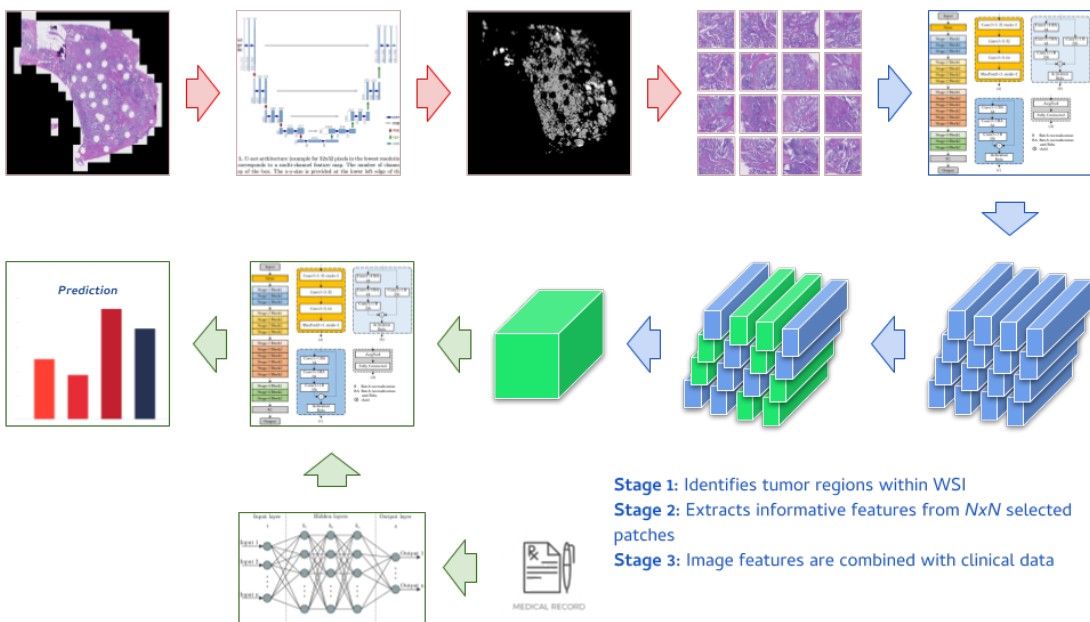

Figure 1: Overview of the proposed three staged multimodal deep learning framework

## 2. Methodology

Our outcome prediction framework employs a three-stage methodology. We trained our models using AdamW (Loshchilov and Hutter, 2019) with Lookahead (Zhang et al., 2019) for optimization and a cosine annealing schedule (Loshchilov and Hutter, 2017) with a warmup phase for the learning rate.

Tumor Detection: The first stage involves precise tumor detection in whole slide images (WSIs) using a custom-designed UNet (Ronneberger et al., 2015) model. The UNet's encoder utilizes ConvNeXtv2 (Woo et al., 2023) for robust feature extraction for tumor segmentation. The decoder combines residual connections, a bottleneck architecture, and Squeeze-and-Excitation (SE) blocks (Hu et al., 2018) to refine segmentation.

Patch-Level Feature Extraction: Following tumor detection, patches are extracted from WSIs. WSIs are divided into 256x256x3 patches at 10x magnification. Patches containing less than 10% tumor pixels are filtered out. During training, at each iteration, we randomly select NxN features from the 2xNxN patch representations generated in the first stage. This random selection serves as an effective augmentation strategy, enabling a WSI to be represented in numerous ways, thereby increasing the diversity and difficulty of the training data enhancing the model's ability to generalize. Selected patches are processed by ResNeXt50 (Xie et al., 2017) pre-trained with DINO (Caron et al., 2021). Translation/color augmentations, Gaussian noise, and pixel dropout are applied to each patch randomly and independently. Then ResNeXt50 SSL weights are used only for patch-level feature extraction compressing 256x256x3 patches into 8x8x2048 representations.

Outcome Prediction: In the third stage, patch-level feature representations from Stage 2 are aggregated to represent the WSI. The features are arranged by tumor ratio (highest

tumor percentage patch representation is the first block). ResNet18 (He et al., 2016) is used as the classifier. This ResNet18 network is modified to work on 8x8x2048 'images/tensors' from stage 2 and to accept clinical parameters. Clinical data is processed by a 2-layer fully connected network, concatenated with ResNet18 output, and passed through a fully connected layer for metastasis classification logits.

We employ ConvNeXt-V2 in the UNet encoder for robust feature extraction, ResNeXt50 pre-trained with DINO for efficient patch-level representation learning, and ResNet18 for efficient outcome prediction with a smaller network size.

## 3. Results

We evaluated our novel multimodal deep learning framework on three outcome prediction tasks, utilizing datasets of varying sizes: biopsy Gleason Grade classification (large dataset), metastasis prediction in prostate cancer using RP WSIs (medium-sized dataset), and BRCA2 mutation prediction (small dataset). For biopsy Gleason Grade classification, our framework achieved a QWK of 0.8240, outperforming the MIL baseline (0.7339). In metastasis prediction, our framework outperformed MIL in WSI-only analysis (AUC: 0.8209 vs. 0.7727), with multimodal analysis further improving performance (AUC: 0.8807). Similarly, for BRCA2 mutation prediction, our framework demonstrated superior performance compared to MIL (AUC: 0.9571 vs. 0.8909).

Table 1: Performance achieved by the proposed framework and the MIL baseline

|  | Biopsy Gleason Grade Classification | Metastasis Prediction | | BRCA2 Mutation Prediction |
|---|---|---|---|---|
|  | WSI | WSI | WSI + Clinical | WSI |
| Metric | QWK | AUC | AUC | AUC |
| #Patients | 5160 | 340 | 340 | 17 |
| MIL | 0.7339 | 0.7727 | 0.7867 | 0.8909 |
| Our Framework | 0.8240 | 0.8209 | 0.8807 | 0.9571 |

## 4. Conclusion

Our novel and computationally efficient three-stage multimodal deep learning framework significantly improves histopathology outcome prediction. By innovatively integrating WSI and clinical data through a tailored pipeline featuring a UNet for tumor detection, strategic patch selection, and DINO pre-trained feature extraction, and an option for efficient fusion of image features with clinical parameters, our method achieved superior performance on Gleason classification, metastasis prediction, and BRCA2 mutation prediction compared to MIL. The consistent performance gains achieved by our novel multimodal integration strategy highlight its significant potential for accurate and robust outcome prediction, offering a promising tool for clinical decision-making and personalized cancer treatment strategies.

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
