# OpenReview forum: "Outcome Prediction in Histopathology: A Multimodal Three-Stage Deep Learning Framework"
_MIDL.io/2025/Short_Papers — MIDL 2025 - Short Papers_

### Official Review · Reviewer_EkMJ · 2025-04-16

**Rating:** 3
**Confidence:** 4

**Summary:**

This paper presents a multimodal three-stage deep learning pipeline for outcome prediction from histopathology whole slide images (WSIs) and optionally clinical data. The framework includes (1) a customized UNet with ConvNeXt-v2 encoder for tumor segmentation, (2) DINO-pretrained ResNeXt50 for feature extraction from tumor-rich patches, and (3) a modified ResNet18 for outcome classification, optionally integrating clinical data. The method outperforms a MIL baseline across three tasks: Gleason grade classification, prostate cancer metastasis prediction, and BRCA2 mutation prediction.

**Strengths:**

- The modular three-stage design is logically organized and leverages state-of-the-art components for segmentation and representation learning.
- Effectively combines image and clinical data for improved predictive power, demonstrating tangible performance benefits.
- Achieves significant improvements over MIL baselines in all tasks, including on low-data scenarios (e.g., BRCA2 prediction).
- The tumor-guided and randomized patch selection adds augmentation-like diversity, improving generalization.
- Incorporating DINO-pretrained weights for feature extraction is well-motivated.

**Weaknesses:**

- The paper does not specify which clinical parameters are used or how missing data are handled.
- It is unclear how much each stage or component (e.g., DINO pretraining, clinical input) contributes individually to performance gains.
- With only 17 patients, performance gains in BRCA2 mutation prediction may not generalize.
- The segmentation-then-classification paradigm may still face computational bottlenecks on very large WSI datasets without stronger discussion of efficiency trade-offs.

---

### Decision · Program_Chairs · 2025-05-01

Accept